# Exploring Knowledge Boundaries in Large Language Models for Retrieval Judgment

## Abstract

Large Language Models (LLMs) are increasingly recognized for their practical applications. However, these models often encounter challenges in dynamically changing knowledge, as well as in managing unknown static knowledge. Retrieval-Augmented Generation (RAG) tackles this challenge and has shown a significant impact on LLMs. Actually, we find that the impact of RAG on the question answering capabilities of LLMs can be categorized into three groups: beneficial, neutral, and harmful. By minimizing retrieval requests that yield neutral or harmful results, we can effectively reduce both time and computational costs, while also improving the overall performance of LLMs. This insight motivates us to differentiate between types of questions using certain metrics as indicators, to decrease the retrieval ratio without compromising performance. In our work, we propose a method that is able to identify different types of questions from this view by training a Knowledge Boundary Model (KBM). Experiments conducted on 11 English and Chinese datasets illustrate that the KBM effectively delineates the knowledge boundary, significantly decreasing the proportion of retrievals required for optimal end-to-end performance. Specifically, we evaluate the effectiveness of KBM in three complex scenarios: dynamic knowledge, long-tail static knowledge, and multi-hop problems, as well as its functionality as an external LLM plug-in.

## 1 Introduction

As Large Language Models (LLMs) evolve, their real-world applications expand, yet they often struggle with dynamically changing and unknown static knowledge, leading to inaccuracies or *hallucinations* (Rawte et al., 2023). Retrieval-Augmented Generation (RAG) effectively addresses these challenges by retrieving relevant external information in real time, enhancing LLMs' ability to provide accurate responses. While RAG can significantly boost performance, it also incurs costs, such as increased retrieval requests and longer response times. This raises a crucial question: when is retrieval truly necessary?

Previous studies on the necessity of RAG for LLMs can be categorized into two main approaches. The first focuses on the *query* itself, with methods like SELF-RAG (Asai et al., 2023) instructing models such as GPT-4 (Achiam et al., 2023) to assess whether retrieving external documents (e.g., Wikipedia) can produce better responses. Although this approach based on query instructing can identify questions that require real-time information, it remains model-agnostic and struggles to determine whether an LLM has mastered specific knowledge. The second approach evaluates both *questions* and *model responses* to determine if an LLM can answer a question, generating data by sampling multiple model responses and using manual labels for evaluation. However, this method is labor-intensive and relies heavily on manual labeling, which can create biases and lead to increased training costs. Recent advancements in adaptive retrieval (Jeong et al., 2024) highlight the importance of tailoring the retrieval process based on the complexity of the questions. This approach improves alignment between user intent and retrieved content by considering the model's responses alongside RAG effects. However, measuring the effectiveness of retrieved information on model outputs remains challenging, especially in complex and open-domain scenarios.

Given these challenges, using GPT-4-based classification labels for questions to generate data for Llama-2 7B evaluations is not reasonable, as it does not consider the knowledge boundaries of Llama-2 7B. In our analysis, we find that performance varies significantly among LLMs of different sizes,

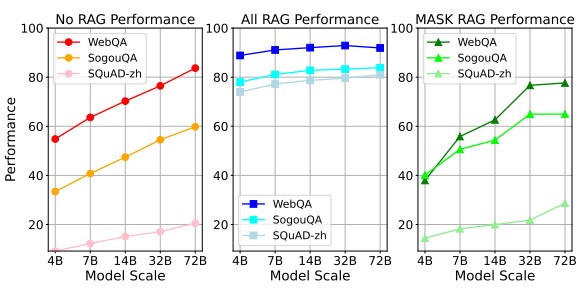

Figure 1: Illustration of the impact of RAG on LLM performance.

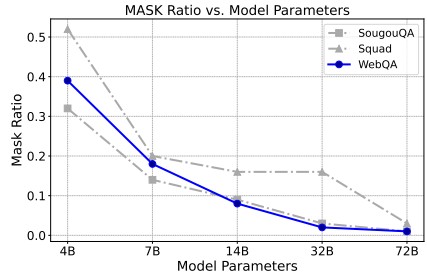

Figure 2: Illustration of the ratio of different LLM output MASK.

indicating the need to focus on specific models for a more accurate simulation of their knowledge boundaries, rather than just distilling larger models for labels. Additionally, generating soft labels from queries and LLM predictions often requires gold answers for evaluation, posing challenges for questions without definitive answers. For questions with gold answers, we sample multiple LLM responses to compute confidence levels. In contrast, for questions lacking gold answers, we evaluate the entropy of phrase distributions from the model's responses to create a certainty distribution. Based on these evaluation indicators, we categorize RAG's impact on LLMs as beneficial, neutral, or harmful based on these evaluation indicators. Our findings show that as the LLM's confidence or certainty increases, the ratio of neutral and harmful queries also rises, while beneficial queries and performance improvements decline. This motivates us to reduce retrieval requests leading to neutral or harmful outcomes, optimizing time and computational efficiency while maximizing beneficial retrievals to enhance question answering (QA) performance.

In this work, we propose methods to evaluate the known and unknown aspects of LLMs based on indices of confidence and certainty derived from sampled responses. By establishing different thresholds for confidence and certainty, we can effectively distinguish between beneficial questions and those deemed neutral or harmful, thereby enabling the generation of data with soft labels. The data is used to fine-tune the Knowledge Boundary Model (KBM) to determine if the LLM considers the question unknown, thus indicating whether retrieval is necessary. This assessment informs decisions on whether RAG enhancement is necessary, leading to better performance and decreased retrieval costs. We evaluate several English and Chinese QA datasets and demonstrate that KBM effectively detects knowledge mastery in LLMs through end-to-end results. Our confidence-based approach significantly reduces retrieval ratios while achieving performance close to ALL RAG, with a 43.17% reduction in retrieval on WebQA and a performance drop of only 1.7%. Similarly, the certainty-based approach maintains performance near ALL RAG (with a 0.42% decrease) and reduces the search ratio by approximately 10% on QA tasks. Further analysis explores KBM's effectiveness in open domains across three dimensions: dynamic knowledge, long-tail static knowledge, and multi-hop problems. Finally, we discuss the impact of incorporating RAG on the knowledge boundaries of LLMs and potential solutions based on KBM to address these challenges.

## 2 RELATED WORK

**Large Language Model Knowledge Exploration**. The exploration of knowledge boundaries in LLMs attracts significant attention. Kadavath et al. (2022) examines the self-evaluation capabilities of LLMs, showing that larger models enhance their calibration by initially proposing answers and then evaluating their validity.However, their model still face challenges in accurately identifying errors. Ren et al. (2023) studies LLMs' perception of factual knowledge boundaries and finds that they often display blind confidence in their abilities. Using retrieval-augmented questioning, the study shows that retrieval can enhance judgment capabilities but highlights a dependence on the quality of supporting documents. Yin et al. (2023) focuses on self-awareness, demonstrating that while LLMs can identify some unanswered questions, substantial discrepancies still exist, affecting their uncertainty detection. Chen et al. (2024) introduces COKE, an unsupervised method for teaching models to articulate their knowledge limits through internal signals, yielding improved outcomes across various datasets. Kang et al. (2024) points out that LLMs often default to examples in training

data when facing unfamiliar queries. This highlights the need for better factuality controls to reduce instances of hallucination. Li et al. (2024b) explores hallucinations related to insufficient prompt context, showing that models frequently fail to recognize inadequate information. They propose *uncertainty-sensitive tuning* to improve response reliability. Based on the above research work, our research introduces the KBM. This model aims to enhance the ability of LLMs to determine when and what external knowledge to retrieve for open-domain QA, revealing the key dynamics between internal and external knowledge utilization.

**Retrieval-Augmented Generation.** RAG enhances LLMs by integrating retrieved text passages, significantly improving performance in knowledge-intensive tasks. A key focus is optimizing the timing and strategy of retrieval. Asai et al. (2023) introduce SELF-RAG, a method that trains LLMs to retrieve information, generate content, and evaluate their outputs using reflection tokens. This method enables the customization of model behavior, demonstrating significant performance improvements over standard RAG approaches. Jeong et al. (2024) proposes Adaptive-RAG, which adjusts query handling based on complexity. The model employ different strategies, including non-retrieval and multi-step approaches,to enhance the relevance of responses to diverse queries. Wu et al. (2024) explores how LLMs process erroneous retrieved content. By creating a dataset to assess model responses to incorrect information, the study reveals insights into how models correct their outputs or may perpetuate errors. Cuconasu et al. (2024) conducts a comprehensive study on the retriever's function in RAG. Key findings reveal that the positioning of relevant documents affects model performance, and unexpectedly, noisy documents can enhance accuracy when placed strategically. These studies demonstrate the ongoing development of RAG methodologies, laying the groundwork for our investigation into new strategies for effective retrieval during generation.

## 3 PRELIMINARIES: LLM KNOWLEDGE BOUNDARIES AND RAG ANALYSIS

This section examines RAG's impact on LLM performance. We first assess LLMs with varying parameter counts on QA tasks , revealing differences in knowledge boundaries related to parameter size. Next, we evaluate how each LLM utilizes textual information retrieved by naive RAG and investigate the effects of MASK perturbation to understand their reliance on RAG. We categorize questions into three types based on RAG's influence on LLM outputs. Additionally, we propose using confidence based on LLM accuracy and the certainty of output words as indicators, motivating further study into simulating knowledge boundaries with these evaluation methods.

### 3.1 HOW DOES RAG AFFECT THE ACCURACY AND UNCERTAINTY OF LLM RESPONSE?

We assess the impact of RAG on the performance of LLMs with varying parameter sizes. Our findings indicate that while LLMs exhibit different abilities to answer questions and possess distinct knowledge boundaries, their capacity to utilize retrieved text information remains largely consistent. Our analysis employs several configurations: LLM Only: This configuration generates responses directly from the LLM. ALL RAG: We enhance the Naive RAG approach by concatenating the top ten blocks retrieved from Google as contextual information. MASK RAG: This variant replaces the gold answers present in RAG with MASK, supplying these modified data as context for the LLM.

As illustrated in Figure 1, we focus on the Qwen1.5 models (4B, 7B, 14B, 32B) alongside the Qwen-2.0 72B model. The evaluation datasets utilized for assessing short question answering and reading comprehension tasks include WebQA (Chang et al., 2022), SogouQA[1], and SQuAD1.5-zh[2]. Our results demonstrate that LLMs improve their performance with increasing parameter sizes across the three datasets, exhibiting a gradual differentiation in their QA capabilities. Notably, when employing Naive RAG, all models, particularly those with 14B-72B parameters, display a strong and consistent ability to exploit contextual information. We find that RAG has a more pronounced effect on smaller LLMs, although the upper limit of performance improvement remains similar across model sizes. For instance, in the WebQA dataset, the accuracy difference between the 7B and 72B models when using RAG is only 3.05%. In contrast, without RAG, this difference escalates to 28.87%. Interestingly, the use of MASK RAG appears to diminish the advantages associated with RAG, potentially leading to

---

[1]https://github.com/sherlcok314159/ChineseMRC-Data

[2]https://github.com/pluto-junzeng/ChineseSquad

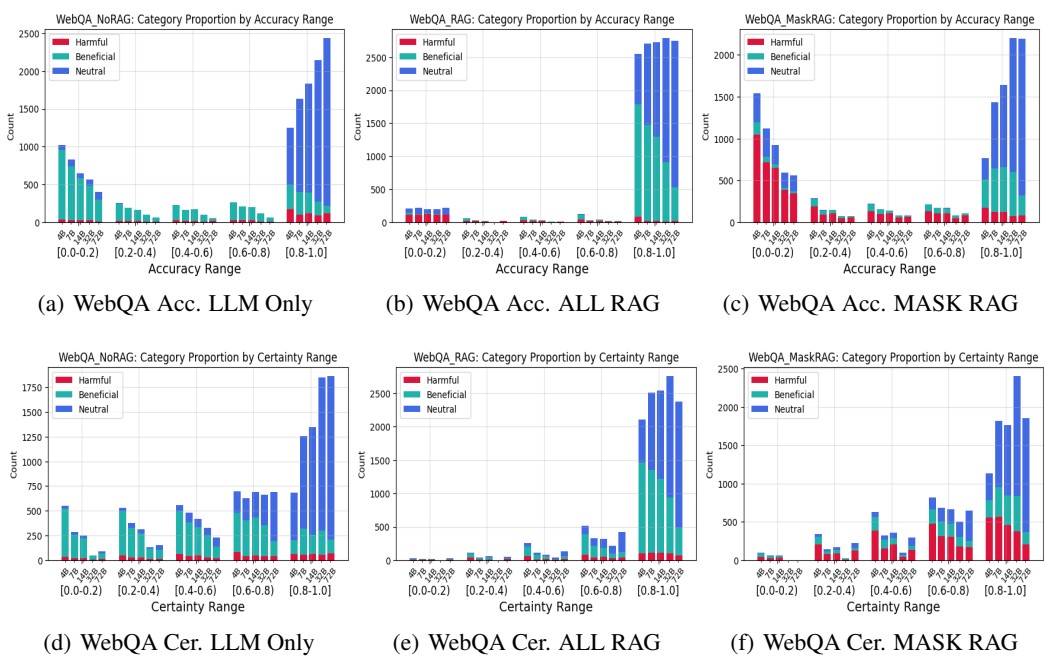

(a) WebQA Acc. LLM Only  (b) WebQA Acc. ALL RAG  (c) WebQA Acc. MASK RAG

(d) WebQA Cer. LLM Only  (e) WebQA Cer. ALL RAG  (f) WebQA Cer. MASK RAG

Figure 3: Predictions for three categories: Beneficial, Neutral, and Harmful on the WebQA test set. The top panel shows accuracy distribution; the bottom panel shows certainty distribution. The left section, LLM Only, shows scores for the three categories. The middle section, ALL RAG, presents scores from the RAG approach. The right section displays scores for MASK RAG.

performance degradation, especially on simpler datasets. The presence of noisy information adversely impacts smaller models to a greater extent.

These findings suggest that different LLMs possess varying knowledge boundaries in question answering and demonstrate distinct retrieval strategies. Although all models exhibit strong capabilities in leveraging context, their resilience to noise interference varies significantly.

## 3.2 THREE DIMENSIONS TO DIVIDE THE BOUNDARIES OF RAG

To more precisely evaluate the impact of RAG on LLM generation, we categorize its effects into three aspects:

- Beneficial: RAG effectively solves this problem.
- Neutral: RAG does not impact the LLM's effectiveness.
- Harmful: RAG compromises the LLM's effectiveness.

To isolate the influences of other modules, we utilize a simplified RAG pipeline for our analysis. Specifically, we leverage Google Open Search to retrieve the top 10 blocks, which are then provided to the LLM. Acknowledging that a single model response may lack robustness or representativeness, we implement a confidence-based categorization method. Inspired by Kadavath et al. (2022), we generate $I = 30$ answer samples at $T = 1$. For a given question $Q$, if 20 of the answers sampled by the model are correct and 10 are wrong, unlike Kadavath et al. (2022) which contains 20 copies of $(Q, M = 1)$ data points and 10 copies of $(Q, M = 0)$ data points. Instead, we construct a confidence of knowledge for this question based on our samples, resulting in a single data point $(Q, M_{pred} = \frac{20}{30})$. This method aims to accurately represent the model's understanding and misconceptions while significantly reducing the size of the training and test datasets by a factor of $I$. Consequently, we approximate the model's soft labels for knowledge using hard labels derived from a diverse set of QA data points.

However, this method becomes challenging in the absence of gold-standard answers. To address this, we simulate the effect based on the uncertainty of the generated responses. We compute the

word/phrase entropy distribution of the LLM based on the 30 generated answers to assess the model's certainty. Let $k$ represent the number of distinct answer types, denoted as $K_1, K_2, ..., K_k$. The probability of each answer type occurring is represented as $P_1, P_2, ..., P_k$, satisfying the normalization condition: $\sum_{i=1}^{k} P_i = 1$. Using this probability distribution, we quantify certainty through entropy with the following formula:

$$C(Q) = 1 - U_{PE}(Q) = 1 + \sum_{i=1}^{k} P_i, \log P_i. \tag{1}$$

where $U_{PE} \in [0, 1]$ represents the entropy value, which quantifies the model's uncertainty or confidence in its answers.

We classify the effects of RAG based on the three categories and two indicators as follows: an effect is deemed **Beneficial** if the indicator increases after incorporating RAG compared to the LLM Only scenario; **Neutral** if the indicator remains unchanged and aligns with the value from the LLM Only condition; and **Harmful** if the indicator decreases following the addition of RAG, falling below the value observed in the LLM Only scenario. Using these categories, we analyze the class distribution of LLMs across accuracy intervals on the WebQA and SogouQA datasets, as shown in Figure 3. The figure indicates that the [0-0.8) interval has the highest proportion of beneficial cases, while neutral and harmful cases peak in the [0.8-1.0] range. This suggests a cost-benefit relationship: higher accuracy reflects greater confidence in answers, reducing the advantages of RAG, especially in the [0.8-1.0] range, where harmful cases also concentrate.

When an LLM displays high confidence, we infer mastery of the solution; low confidence implies a lack of understanding. This assessment combines human annotations, model predictions, and evaluation methods. The uncertainty metric highlights components such as Aleatoric and Model Uncertainty. Low uncertainty corresponds to high confidence, while high uncertainty indicates doubt. Thus, determining mastery can significantly lower retrieval needs, as illustrated in the blue area of the figure. For instance, with mastery defined as accuracy exceeding 0.8, a 7B model on WebQA reduces retrieval by 55.47% and enhances performance by 0.92%. A 72B model achieves an 80.95% reduction in retrieval and a 2.52% performance gain. If we set a confidence threshold of under 0.2, the 7B model reduces retrieval by 41.45%, but this results in a 4.15% performance decline compared to AllRAG. The 72B model achieves a 61.56% reduction with a 1.56% drop in performance. We observe a Pearson correlation coefficient of 0.64 between confidence and certainty, indicating a positive relationship. However, certainty-based metrics tend to be more inclusive.

Overall, the thresholds for mastery and confidence depend on the specific dataset and model. While accuracy aligns with human intent and reflects model capabilities, its reliability is affected by annotation quality and evaluation variability. Uncertainty metrics also have limitations, particularly concerning model overconfidence, which can degrade performance. These aspects will be further examined in the following sections.

## 3.3 THE DEPENDENCE OF LLM ON RAG

In this section, we explore the reliance on RAG from the perspective of LLM uncertainty. We find that incorporating RAG significantly enhances LLM certainty, which is evident in the greater consistency of the generated answers. This observation raises a critical question: does the model primarily copy answers from the RAG input to reduce uncertainty, or does it absorb knowledge from the provided material? To address this question, we conduct an experiment with scenarios where the model outputs exhibit a certainty of 1, indicating high confidence in predictions. As shown in Figure 2, we measure the frequency of outputs corresponding to MASK under MASK RAG. If the LLM mainly copies answers from the RAG input, we expect to observe a high proportion of MASK outputs; this would imply that the model retrieves and reproduces answers rather than inferring them from context. Our findings are consistent across WebQA, SogouQA and SQuAD-zh datasets. Notably, the tendency to copy answers decreases with an increase in the number of model parameters, leading to a lower proportion of MASK outputs. This suggests that larger models are more adept at leveraging non-answer content to derive predictions, rather than relying on direct answer extraction. Such reliance demands more from RAG and may negatively affect LLM performance.

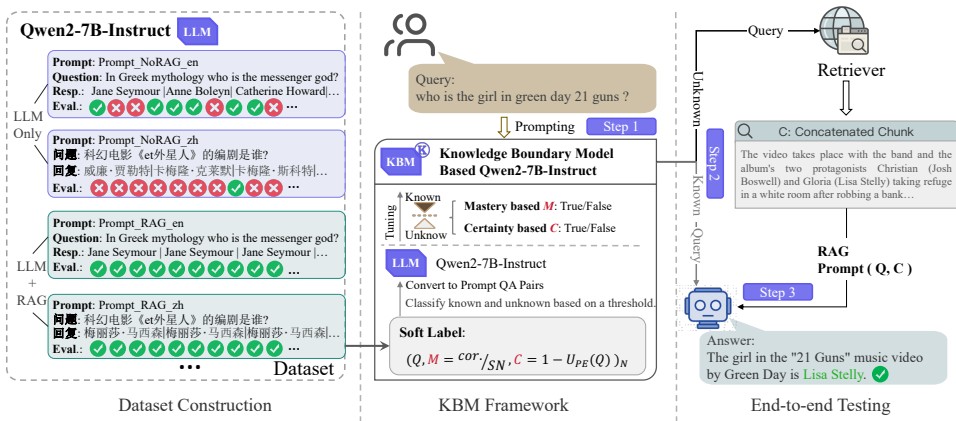

Figure 4: Illustration of the workflow for data generation, model training, and inference processes.

# 4 CAN LLM EXPRESS KNOW/UNKNOWN?

In Section §3.2, we noted that using confidence and certainty as screening indicators helps identify questions with high mastery and low confidence for retrieval enhancement. This approach can reduce the proportion of retrieval and provide performance benefits. So, how can we allow LLMs to express their mastery and certainty? We explore two simple approaches to examine two key questions: Can the model effectively express mastery and certainty? Are there internal model features or data distribution characteristics related to knowledge that the model does not master or finds uncertain?

## 4.1 METHODS

To construct training data for the KBM, we generate soft labels using the sampling method detailed in Section §3.2. As shown in Figure 4, each query is assessed under two configurations: LLM Only and RAG. Based on accuracy and certainty, we derive two data fields: *mastery* and *certainty*. We establish a threshold, $\tau$, to differentiate between known and unknown data, with values below $\tau$ labeled as False and those above $\tau$ labeled as True. In the training phase, the generated data is formatted into question-answer pairs and utilized to fine-tune the sampled ontology model (Qwen2-7B instruct). Further details regarding the settings can be found in the Appendix §A.1.

During the inference phase of the KBM, the process unfolds in three steps: At **step 1**, after the user enters a query, the system packages this query through a prompt and sends it to the KBM to assess mastery and certainty. At **step 2**, if the KBM judges the query as True, it forwards the original query directly to the answer generation model (Qwen2-7B instruct) to generate a response. If the query is judged as False, the process proceeds to step 3. At **step 3**, the system performs an open-domain retrieval using Google, based on the user's original query. The retrieved chunks of information are then spliced together. The original query and the connected chunks are combined using RAG Prompt and subsequently fed into the answer generation model to produce the final response.

## 4.2 BASELINES AND DATASET

**Baselines and Metrics**. We establish a baseline using the following methods: (1) **Prompting Methodology**: We utilize state-of-the-art models, including LLAMA3-70B, Qwen2-72B, and GPT-4. Each model is presented with the same retrieval query. If a model indicates a need for additional information, we perform a Google search and enhance the response using the top 10 results through RAG. If no retrieval is necessary, the model generates the answer autonomously. (2) **Random**: This method serves as a dynamic benchmark, involving random extractions of queries for RAG. The search proportions align with the retrieval ratios established by each baseline method. Each methodology offers a unique approach to managing knowledge boundaries, contributing to a comprehensive evaluation of our proposed frameworks. In terms of performance **Metrics**, we use Exact Match (EM) for NaturalQA and TrivialQA, and Accuracy for the other datasets, while also considering the retrieval ratio (Rat.) as an important metric. Further details can be found in the Appendix §A.2.

| | A/E | Mastery Rat(%) | Rnd | A/E | Certainty Rat(%) | Rnd | A/E | Prompt Rat(%) | Rnd | LLM Only | All RAG |
|---|---|---|---|---|---|---|---|---|---|---|---|
| *KBM (Qwen2-7B-Instruct based)* | | | | | | | | | | | |
| en-NaturalQA | 60.09 | 86.15% | 51.20 | **61.27** | 97.53% | 55.16 | 59.92 | 85.98% | 57.91 | 35.58 | 61.63 |
| en-TirviaQA | 81.39 | 70.25% | 74.42 | **83.32** | 87.70% | 80.66 | 79.08 | 73.55% | 75.62 | 50.79 | 84.60 |
| en-OpenBoook | **82.64** | 26.20% | 81.92 | 82.32 | 57.80% | 82.88 | 81.96 | 27.80% | 82.12 | 82.72 | 81.48 |
| en-MMLU | 70.00 | 70.01% | 68.59 | **70.45** | 65.31% | 69.14 | 67.30 | 27.24% | 67.70 | 66.45 | 70.80 |
| en-SQuAD | 89.13 | 87.18% | 83.14 | **89.38** | 97.27% | 88.61 | 78.55 | 75.55% | 75.30 | 34.52 | 89.83 |
| en-FreshQA | 58.40 | 86.80% | 57.80 | **59.72** | 91.80% | 58.56 | 58.52 | 86.80% | 57.82 | 33.68 | 60.84 |
| zh-WebQA | 90.84 | 56.83% | 81.19 | **92.12** | 89.71% | 90.71 | 89.75 | 77.47% | 87.01 | 70.06 | 92.54 |
| zh-SogouQA | 81.59 | 75.23% | 74.17 | **82.89** | 94.56% | 81.06 | 78.13 | 78.90% | 76.75 | 52.53 | 83.01 |
| zh-SQuAD | 82.66 | 96.00% | 81.29 | **83.45** | 99.33% | 83.31 | 78.74 | 90.33% | 77.82 | 22.72 | 83.66 |
| zh-CMMLU | **78.32** | 49.98% | 77.73 | 77.12 | 95.29% | 77.06 | 77.97 | 49.71% | 77.87 | 78.35 | 77.05 |
| zh-Ceval | **73.56** | 52.11% | 71.99 | 71.71 | 94.03% | 71.69 | 72.89 | 51.28% | 72.39 | 72.93 | 71.62 |

Table 1: Evaluation results of KBM on 11 test sets. LLM Only refers to baselines operating without retrieval, while ALL RAG represents baselines utilizing RAG.

**Dataset**. NaturalQA (Kwiatkowski et al., 2019): This QA dataset, curated by Google, consists of real-world questions derived from natural retrieval queries. TriviaQA (Joshi et al., 2017): This dataset is based on encyclopedic content and features complex questions and answers, primarily sourced from competitions and quizzes. MMLU (Hendrycks et al., 2021): This dataset comprises multiple-choice questions across various fields, assessing the model's knowledge mastery in academic and professional domains. OpenBookQA (Mihaylov et al., 2018): Focusing on scientific inquiries, this dataset requires reasoning rooted in principles and common sense. en-SQuAD-en2.0 (Rajpurkar et al., 2018): This dataset features question-answer pairs and evaluates reading comprehension skills. FreshQA-en (Vu et al., 2023): This dataset presents various question and answer types, offering a comprehensive assessment of QA capabilities. HotpotQA (Yang et al., 2018): This dataset consists of 113,000 Wikipedia based QA pairs that necessitate complex reasoning across multiple supporting documents and include sentence-level supporting facts. For the Chinese dataset, we utilize the following resources: WebQA (Chang et al., 2022): This open-domain QA dataset is collected via web crawlers, covering a wide array of topics and evaluating the performance of QA systems. SogouQA [3]: Provided by Sogou, this dataset features user-generated questions and system-generated answers, assessing accuracy and robustness. MLEC (Li et al., 2021): This dataset is designed to test the comprehension capabilities of models in various contexts. Xiezhi (Gu et al., 2024): A set of 249,587 Chinese/English questions covering 516 subjects for evaluating LLMs. SQuAD-zh [4]: The Chinese version of the English SQuAD (Rajpurkar et al., 2018) dataset serves to train and evaluate machine reading comprehension and QA systems. C-EVAL (Huang et al., 2023): This comprehensive Chinese evaluation suite assesses large language models' knowledge and reasoning across 52 disciplines and four difficulty levels. CMMLU (Li et al., 2024a): This comprehensive benchmark assesses the knowledge and reasoning capabilities of language models across 67 topics, from basic to advanced. We used WebQA, TriviaQA, and a combination of MMLU, MLEC, and XieZhi training sets to train the KBM model. The test set and specific data information are shown in the Appendix.

## 5 SIMULATE KNOWLEDGE BOUNDARY

### 5.1 END-TO-END EVALUATION

We evaluate the end-to-end performance of KBM across 11 test sets, demonstrating that both Mastery and Certainty reduce the search ratio while enhancing the LLM's ability to answer questions. Mastery adopts a more conservative approach to search ratio management, while Certainty emphasizes performance improvement. The results are presented in Table 1.

Our analysis reveals that for short answer and QA test sets, such as en-NaturalQA, en-TriviaQA, zh-WebQA, and zh-SogouQA, both methods outperform the random baseline, with Mastery achieving

---

[3]https://github.com/sherlcok314159/ChineseMRC-Data
[4]https://github.com/pluto-junzeng/ChineseSquad

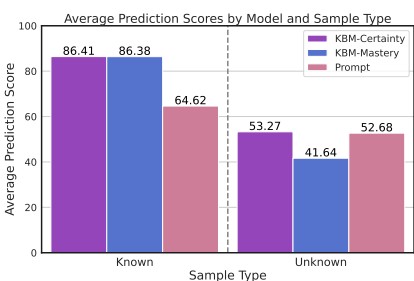

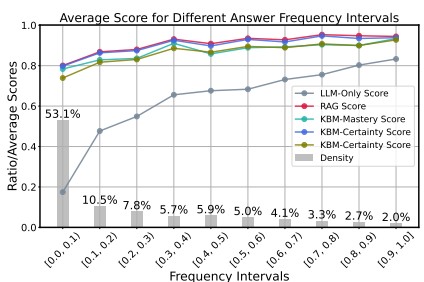

Figure 5: The performance of LLM in the case of KBM model being judged as Known/Unknown.

Figure 6: Comparison of performance changes under different knowledge frequencies.

particularly notable results. In reading comprehension tasks like en-SQuAD and zh-SQuAD, where background information is crucial for accurate answers, KBM exhibits a higher search ratio. All methods demonstrate strong performance and a high search ratio on en-FreshQA, indicating that KBM and LLM effectively capture queries with temporal features. However, in the context of multiple-choice questions found in CMMLU, MMLU, and OpenBookQA, our method shows only marginal improvements over the random baseline. We suspect this limited enhancement arises from the challenges in sourcing relevant information for multiple-choice formats.

KBM enables the model to effectively distinguish between known and unknown information, thereby optimizing performance and reducing costs. We analyze the mean scores across all test sets when categorized as known or unknown, as illustrated in Figure 5. The results indicate that when classified as known, the mean answer score of the LLM is higher, while the score decreases when information is deemed unknown. This pattern aligns with our expectations regarding the differentiation process. In contrast, although the prompt-based method can distinguish between known and unknown information, its effectiveness is comparatively lower.

| | fast-changing | | slow-changing | | never-changing | |
| | Ratio | Acc. | Ratio | Acc. | Ratio | Acc. |
| --- | --- | --- | --- | --- | --- | --- |
| *KBM* | | | | | | |
| -Mastery | 94.6% | 51.6 | 92.3% | 59.0 | 75.9% | 62.7 |
| -Certainty | 98.5% | **52.0** | 95.6% | **60.2** | 89.4% | **64.6** |
| Prompt | 93.8% | 51.5 | 86.8% | 57.8 | 81.8% | 64.1 |

| | en-FreshQA($\geq$2hop) | | HotpotQA | |
| | Ratio | Acc. | Ratio | Acc. |
| --- | --- | --- | --- | --- |
| *KBM* | | | | |
| -Mastery | 96.5% | **48.9** | 91.2% | 50.9 |
| -Certainty | 99.1% | 48.7 | 94.3% | **51.4** |
| Prompt | 91.3% | 48.3 | 93.7% | 51.2 |

Table 2: Retrieval judgment results of dynamic changing class tests.

Table 3: Retrieval judgment results of multi-hop class tests.

## 5.2 ANALYSIS

**Dynamic Knowledge: Utilizing KBM for Identifying Questions with Temporal Variability**. We demonstrate that KBM effectively identifies questions with answers that change over time. Specifically, we classify the temporal changes in answers found in the FreshQA dataset into three distinct categories: fast-changing, slow-changing, and never-changing, based on the frequency of these changes. In open domains, variations in answers often signal the need for knowledge updates, necessitating the integration of external information into the LLM. As illustrated in Table 2, the end-to-end performance without RAG for both fast-changing and slow-changing categories is suboptimal, indicating a reliance on external knowledge. The high retrieval rates of KBM for both fast-changing and slow-changing categories suggest that it adeptly captures these evolving answers. Conversely, the lower retrieval rate for the never-changing category implies that some knowledge is effectively embedded within the LLM. This observation underscores KBM's sensitivity in identifying questions with temporally correlated answers, highlighting its role in enhancing dynamic knowledge adaptation.

**Long-tail Static Knowledge: Evaluating KBM's Performance Across Knowledge Frequencies**. We investigate the capacity of KBM to capture low-frequency long-tail knowledge across various question sets. Specifically, we combine test data from WebQA, SogouQA, and SQuAD-zh. Utilizing

| | KBM-Mastery | | | KBM-Certainty | | | Prompt | | | LLM Only | ALL RAG |
|---|---|---|---|---|---|---|---|---|---|---|---|
| | A/E | Rat(%) | Rnd | A/E | Rat(%) | Rnd | A/E | Rat(%) | Rnd | A/E | A/E |
| *KBM For zh-WebQA* | | | | | | | | | | | |
| -GPT-4 | 85.7 | 57% | 78.5 | 89.5 | 90% | 87.8 | 69.5 | 14% | 68.0 | 64.3 | 90.0 |
| -Qwen2-72B | 91.4 | 57% | 88.2 | **91.8** | 90% | 91.3 | 87.9 | 22% | 85.6 | 83.7 | 91.9 |
| -Llama3-70B | 77.9 | 57% | 72.6 | 88.2 | 90% | 86.7 | 59.7 | 16% | 55.3 | 48.7 | 90.9 |
| *KBM For en-NaturalQA* | | | | | | | | | | | |
| -GPT-4o | 60.6 | 86% | 60.4 | **60.7** | 98% | 60.5 | 55.7 | 18% | 56.6 | 55.6 | 60.1 |
| -Qwen2-72B | 58.5 | 86% | 56.5 | 59.3 | 98% | 58.9 | 54.7 | 51% | 52.6 | 42.7 | 59.4 |
| -Llama3-70B | 59.6 | 86% | 58.6 | 60.2 | 98% | 59.7 | 59.2 | 47% | 58.6 | 47.2 | 60.2 |

Table 4: End-to-end result with KBM as a plug-in for various LLMs' retrieval judgment modules.

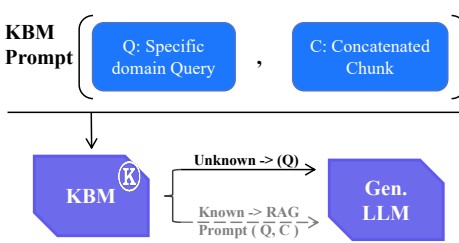

Figure 7: Illustration of KBM exploring the boundaries of RAG.

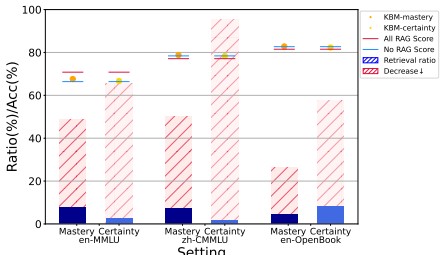

Figure 8: Comparison of KBM double judgment results under RAG.

the gold answers from these datasets, we conduct vector retrieval within our Chinese database to differentiate knowledge based on its frequency. As illustrated in Figure 6, the LLM Only approach demonstrates reduced accuracy for low-frequency knowledge answers while performing better for high-frequency knowledge. However, integrating KBM with a prompt-based retrieval mechanism significantly enhances the model's performance for long-tail low-frequency knowledge. Notably, the Certainty-based method yields the most substantial improvement, followed by the Mastery-based approach. These findings indicate that KBM effectively detects low-frequency long-tail knowledge and boosts overall performance through strategic retrieval.

**Multi-Hop: Multi-hop Knowledge Detection with KBM**. A crucial aspect of our analysis is the ability of KBM to detect complex queries that necessitate multi-hop knowledge. Multi-hop questions comprise intricate knowledge components, requiring adjustments to the LLM. In these scenarios, KBM identifies the complexity of the queries and effectively employs RAG. We tested queries involving two or more hops from the FreshQA and Hotpot test sets, with results presented in Table 6. KBM demonstrates higher retrieval rates for multi-hop questions, indicating its effectiveness in navigating these complexities. While the overall end-to-end improvement is modest, this limitation stems from the need for further optimization of the RAG pipeline.

## 5.3 USE AS A PLUG-IN

We conduct an evaluation of several prominent LLMs using two benchmark datasets: WebQA and NaturalQA. For this analysis, we utilize the KBM as the retrieval judgment model and compare its performance against the LLM Only, RAG, and prompt methodologies applied to each LLM. Our findings reveal that KBM acts as a valuable plug-in, enhancing the end-to-end performance of other LLMs, although it falls short of achieving the comprehensive improvements offered by ALL RAG. This observation aligns with our previous analysis, which indicates that each LLM possesses a unique knowledge boundary, making it challenging to accurately represent the knowledge boundaries of all LLMs with a single model. The results are summarized in Table 4.

In particular, RAG demonstrates its capacity to enhance the response quality of LLMs, with a relatively stable upper limit for this improvement. While the effects on the LLM Only model vary

significantly among the different LLMs, the scores converge when RAG is incorporated. The varying direct knowledge boundaries of LLMs contribute to the disparate enhancement effects observed with KBM. For instance, the KBM-Certainty judgment method shows that Qwen2-72B achieves an increase from 83.7% to 91.8%, representing an improvement of 8.1% points. In contrast, Llama3-70B exhibits a more substantial increase, improving from 48.7% to 88.2%, which corresponds to a remarkable gain of 39.5% points.

Overall, compared to the baseline, the knowledge boundary model typically enhances LLM performance across the test datasets. However, discrepancies exist between the known/unknown distributions of the knowledge boundary model and the general model. This divergence accounts for the varied performance enhancements observed among the different LLMs. For example, in robust LLMs, excessive unnecessary retrievals may occur. While Qwen2-72B may demonstrate a lower upper limit for retrieval, the knowledge boundary model still manages to execute 90% of the retrievals. Conversely, Llama3-70B encounters challenges in attaining the ALL RAG score even with a 90% retrieval rate in the WebQA dataset. Similar trends manifest in the English QA tests, suggesting that when the English proficiency of the KBM ontology model is lower than that of the generative LLM, performance improvements are more pronounced, and the reverse holds true as well.

## 5.4 DISCUSSION

In this work, we find that while RAG effectively addresses certain challenges associated with LLMs, it is not universally applicable. As noted in Section §3.3, it can be counterproductive when relevant information is difficult to retrieve or noisy. This limitation is illustrated in Table 1, where RAG shows minimal performance improvement in tasks such as multiple-choice questions and math assessments.

We suggest that RAG's effectiveness could be enhanced when deployed within specific modules through techniques such as question rewriting, page reading, and re-ranking. However, our primary focus is to investigate the knowledge boundaries of LLMs. Therefore, we consider strategies to mitigate the impact of RAG components that are neutral or detrimental, especially when LLM knowledge remains limited, thereby emphasizing RAG's boundaries.

To achieve this objective, we implement the method illustrated in Figure 7, which aims to improve information processing while minimizing the influx of irrelevant input tokens and reducing noise from irrelevant retrieval results. Specifically, in specialized problem domains, such as multiple-choice questions, when the KBM identifies a query as unknown, we perform a search to retrieve concatenated chunks of relevant information. The KBM uses both the original query and the concatenated chunks as prompt input to assess whether the LLM can effectively answer the query with the additional retrieved information. If the LLM generates a response, the concatenated chunk is then forwarded to it; if not, the original query is transmitted directly. This method effectively minimizes the amount of information fed into the LLM (as indicated by the red area in Figure 8), while maintaining performance that is competitive with both LLM Only and ALL RAG scenarios, and in some instances, it even surpasses both approaches.

## 6 CONCLUSION

In this paper, we explore the impact of RAG on LLMs, depending on the types of questions they encounter. We find that a key limitation of existing approaches is that RAG often lacks adaptability to the knowledge boundaries of specific LLMs, which can lead to ineffective or even harmful responses. To address this challenge, we first classify the types of questions encountered by LLMs and identify appropriate search regions within the model boundaries. We then introduce a KBM for retrieval judgment, optimizing retrieval requests based on our proposed confidence and certainty methods. This model effectively classifies questions as known or unknown and determines whether retrieval is necessary. Our extensive evaluation on 11 English and Chinese datasets demonstrates that KBM significantly improves retrieval efficiency while maintaining high performance. It proves effective in three complex scenarios: dynamic knowledge, long-tail static knowledge, and multi-hop questions, showcasing its versatility in real-world applications. Our findings indicate that KBM, as an intelligent retrieval judgment strategy, effectively reduces resource consumption by decreasing retrieval calls, token usage, and response costs, while simultaneously enhancing the performance of LLMs.

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

---

**en–Prompt for Sampling LLM–Only Response:**

"Given the following question, give the concise sentence/phrase/noun/entity as answer:\n"

  "question: {query}\n"

  "answer: "

- - -

**zh–Prompt for Sampling LLM–Only Response:**

"给定以下问题，给出最简洁的短语/名词/实体作为答案:\n"

  "问题: {query}\n"

  "答案: "

---

Figure 9: en/zh- Prompt for Sampling LLM Only Response

---

**en–Prompt for Sampling RAG Response:**

"Here's some background information: {evidence}\n"

  "Given the following question, give the concise sentence/phrase/noun/entity as answer:\n"

  "question: {query}\n"

  "answer: "

- - -

**zh–Prompt for Sampling RAG Response:**

"这里有一些背景资料: {evidence}\n"

  "现在给定以下问题，给出最简洁的短语/名词/实体作为答案:\n"

  "问题: {query}\n"

  "答案: "

---

Figure 10: en/zh- Prompt for Sampling RAG Response

---

**en–Prompt for Multiple Choice Question LLM–Only Response:**

"Given the following multiple choice questions, only output the options corresponding to the correct answers (e.g A/B/C/D...), and do not output other content:\n"

  "question: {query}\n"

  "answer: "

- - -

**zh–Prompt for Multiple Choice Question LLM–Only Response:**

"给定以下多项选择题, 仅输出正确答案所在的选项(A/B/C/D/E):\n"

  "问题: {query}\n"

  "答案: "

---

Figure 11: en/zh- Prompt for Multiple Choice Question LLM Only Response

---

**en–Prompt for Multiple Choice Question RAG Response:**

"Here's some background information: {evidence}\n"

  "Given the following multiple choice questions, only output the options corresponding to the correct answers (e.g A/B/C/D...), and do not output other content:\n"

  "question: {query}\n"

  "answer: "

- - -

**zh–Prompt for Multiple Choice Question RAG Response:**

"这里有一些背景资料: {evidence}\n"

  "给定以下多项选择题, 仅输出正确答案所在的选项(A/B/C/D/E):\n"

  "问题: {query}\n"

  "答案: "

---

Figure 12: en/zh- Prompt for Multiple Choice Question RAG Response

## A  APPENDIX

### A.1  IMPLEMENTATION DETAILS

We employ four Nvidia A100 GPUs, each with 80GB of memory, to train our KBM models. Each model undergoes training for three epochs, utilizing a batch size of four. The peak learning rate is set to 1e-5, with a warmup ratio of 2% and cosine decay for the learning rate. To accommodate memory

---

**zh–Prompt for Response Accuracy Assessment:**

"给定预测和答案，如果预测中含有答案请输出1，不包含则输出0.\n"

 "预测: {pred}\n"

 "答案: {answer}\n"

 "输出: "

---

Figure 13: zh-Prompt for Response Accuracy Assessment

---

**KBM Prompt for English Questions with RAG Demonstration：**

"Please assess your mastery and certainty regarding the question based on the information below.\n"

 "Reference: The Hallé is an English symphony orchestra based in Manchester, England. Since 1996, the orchestra has been resident at the Bridgewater Hall in Manchester. The Free Trade Hall on Peter Street, Manchester, England, was constructed in 1853–56 on St Peter's Fields, the site of the Peterloo Massacre. The Free Trade Hall became home to Manchester's Halle Orchestra. In fact it continued to be their home right up until 1996. That is when the Bridgewater Hall ... Oct 5, 2023 \n"

 "Question: Which Manchester building was home to the Halle Orchestra until 1996? \n"

 "Please respond: Do you fully master this question? (mastery: True/False); Are you confident in your ability to answer this question? (certainty: True/False) \n"

 "Answer : {"mastery": "True/False", "certainty": "True/False"} "

- - - - - - - - - - - - - - - - - - - - - - - - - - - - - - - - - - - - - - - - - - - - - - - - - - - - -

**KBM Prompt for Chinese Questions without RAG Demonstration**

"给定一个问题，评估你在此问题上的熟练度（mastery）以及确定性（certainty）\n"

 "参考资料: None\n"

 "问题: 酒量的大小是由身体的哪一个器官来决定的？\n"

 "请回答: 你是否完全掌握此问题? (mastery: True/False); 你对自己能回答这个问题有信心吗? (certainty: True/False)\n "

 "输出: {"mastery": "True", "certainty": "True"} "

---

Figure 14: KBM Prompt for English/Chinese Questions with/without RAG Demonstration

---

limitations, we restrict the maximum token length to 1580 for the 7B model and 1524 for the 13B model. For multi-GPU distributed training, we utilize Deepspeed stage 2 (Rajbhandari et al., 2020) while enabling Bfloat16 precision. Inference on the trained models is conducted using a single Nvidia Tesla V100 GPU with 32GB of memory.

## A.2 Data Generation and KBM Inference Instructions

In this section, we show the instructions for producing data and KBM inference. Figure 9 shows the instructions for collecting LLM Only responses under QA tasks. Figure 10 shows the instructions for collecting LLM responses under RAG. Similarly, Figure 11 and Figure 12 show the instructions for collecting LLM Only/RAG in answering multiple-choice questions. It is also worth noting that since there are no abundant answers in the Chinese data QA data WebQA and SogouQA, directly using the full match method for evaluation will have a large number of errors, so we use Qwen2-72B and the evaluation prompt shown in Figure 13 for answer evaluation. In order to ensure the unbiasedness of the evaluation, we did not include the original question information in the instructions. During KBM training and inference, we adopt the prompt presented in Figure 14 and illustrate the dictionary reply format based on Mastery and Certainty.

