# OpenReview forum: "Exploring Knowledge Boundaries in Large Language Models for Retrieval Judgment"
_ICLR.cc/2025/Conference — ICLR 2025 Conference Withdrawn Submission_

### Official Review · Reviewer_EMPQ · 2024-11-02

**Soundness:** 3
**Presentation:** 2
**Contribution:** 2
**Rating:** 5
**Confidence:** 3

**Summary:**

The authors identify that the impact of RAG techniques on a given question and LLM falls into three categories: beneficial, neutral, and harmful. By identifying and excluding neutral and harmful RAG cases, retrieval time can be reduced, and the accuracy of LLM-based question answering can be improved. The authors find a high correlation between their proposed accuracy and certainty metrics and whether the use of RAG for a particular question is beneficial, neutral, or harmful.

Based on this finding, the authors train a KBM model that uses the accuracy and certainty metrics to determine whether an LLM should employ RAG techniques. They observe that KBM significantly enhances retrieval efficiency while maintaining high performance.

**Strengths:**

- The problem identified by the authors is intriguing: in certain cases, using RAG techniques does not consistently enhance the performance of LLMs. For some scenarios, it is more advantageous to rely on direct inference by the LLM rather than employing RAG techniques.

- The authors provide extensive experimental results.

**Weaknesses:**

- The terminology definitions in Section 3.2 are somewhat confusing. According to the caption of Figure 3, the two metrics should be accuracy and certainty. However, in lines 238–249, terms like "confidence threshold" and "mastery" appear, which makes it unclear how these terms relate to or differ from the proposed accuracy and certainty metrics.

- The methodology in Section 4.1 is somewhat brief. Based on Section 4.1 and Figure 4, it is unclear how the threshold $\tau$ is applied. Specifically, does $\tau$ assess accuracy and certainty separately to determine their individual true or false states, or do both metrics need to exceed $\tau$ to be considered true? Further clarification would be beneficial.

**Questions:**

- I find Figure 3 (a) and (d) somewhat difficult to understand. According to the definitions provided in lines 229–232, the categories beneficial, neutral, and harmful are defined by comparing the results after incorporating RAG techniques with those from LLM-only. It is unclear what beneficial, neutral, and harmful refer to in the LLM-only panels (a) and (d), as the basis for comparison is not evident.

---

### Official Review · Reviewer_vojs · 2024-11-06

**Soundness:** 2
**Presentation:** 2
**Contribution:** 2
**Rating:** 3
**Confidence:** 4

**Summary:**

This paper examines the effect of the confidence of LLM-generated responses on the RAG’s performance in QA tasks, dividing the impact of RAG on LLM into three categories --- 1) beneficial, 2) neural, and 3) harmful. The work further proposes the knowledge boundary model (KBM), which predicts whether LLM knows the query, so whether RAG is helpful or not. The experiment results show that the KBM-based RAG reduces the ratio of unnecessary retrievals, showing performances close to the original RAG (i.e., ALL RAG).

**Strengths:**

1.	The effect of RAG on QA tasks under various ranges of confidence is examined in a sophisticated and detailed manner, which is helpful and interesting to the literature.
2.	The paper proposes knowledge boundary model (KBM), shows that the resulting KBM-based RAG is promising, improving the efficiency of RAG, while reasonably keeping the original performances.

**Weaknesses:**

1.	The presentation of the paper is not clear, and some technical parts remain unclear, requiring a substantial revision. Here are examples:

- In Eq. (1), why UE, the entropy term, belongs to [0,1]?
- Three categories – Beneficial, Neural, and Harmful – are defined based on the performance gain by RAG over LLM only. But, how Beneficial, Neural, and Harmful types are defined on three settings – LLM only, ALL RAG, and MASK RAG?
- Eq. (1) is computed under LLM only, or RAG?
- In Section 3.2, what is the actual definition of Accuracy in Figure 3?

2.	In the experiments, the proposed KBM-based RAG is reported comparing to RAG. Here, only prompt-based filtering is only used, but given extensive studies on knowledge boundary, more baselines to predict the boundary of knowledge need to be compared. In particular, much simpler baseline could be designed using Eq. (1) with confidence-based decision.

**Questions:**

Please see weaknesses part.

---

### Official Review · Reviewer_hJzJ · 2024-11-06

**Soundness:** 2
**Presentation:** 2
**Contribution:** 3
**Rating:** 5
**Confidence:** 4

**Summary:**

The authors explore the impact of retrieval on the Question Answering (QA) capabilities of LLMs, identifying cases where a retrieval step may be unnecessary or could even harm the performance of retrieval-augmented models. They propose fine-tuning a separate Knowledge Boundary Model (using `Qwen2-7B-Instruct` as base) that would be able to predict the expertise of the *reader* model according to two separate criteria: (i) mastery (i.e. by using the gold labels for indicating the portion of sampled answers that is correct) and (ii) certainty (i.e. by considering probability scores of possible answers). Evaluation experiments are conducted across different datasets involving both English and Chinese QA scenarios.

**Strengths:**

- The paper demonstrates that consistently performing a retrieval step does not necessarily yield performance improvements. In cases where the retrieved passages are suboptimal (i.e. the authors seek to approximate this setting with the MASK RAG setup), retrieval can even lead to a decline in accuracy.
- Experiments are conducted across different datasets involving both English and Chinese QA scenarios.
- Interesting finding about the fact that larger models do not rely on extracting the answer directly from the retrieved content, and that they can derive knowledge by relying more on the parametric knowledge.

**Weaknesses:**

- The motivation for the approach is not fully clear. While the authors aim to reduce computational costs by fine-tuning `Qwen2-7B-Instruct` as a Knowledge Boundary Model, there is no analysis on whether this method would indeed be more efficient than performing a retrieval step at each inference round. The paper would benefit from a comparison of the computational costs associated with using the Knowledge Boundary Model as a mastery or certainty predictor versus additional retrieval steps, which would also potentially include the increased content for the subsequent reader LLM.
- While one of the motivations is to minimise retrieval requests when the retrieved content might be neutral or harmful, it is unclear from the provided experiments how the resulting system would perform under these conditions.
- Lack of strong baselines. The paper does not properly relate its contributions to prior works in this space, and the included baselines are based on rather simple prompting strategies.
- The presentation could be enhanced, as several points throughout the paper lack clarity (please refer to the Questions sections below).
- No significant accuracy improvements using the mastery and certainty thresholds in multi-hop QA scenarios.

**Questions:**

- Why weren’t other systems, such as the one proposed by Asai et al. (2024), which introduce more sophisticated retrieval pipelines within the general RAG framework, included as baselines?
- How are the thresholds $\tau$ for mastery and certainty (i.e. in lines 300-301) tuned?
- In Eq. 1, I understand that $P_i$ can be decomposed into the probabilities of decoding the intermediate tokens of generating answer $i$. Do you normalise these decoding probabilities (i.e. the probability of decoding the final answer by considering the probabilities of its constituent tokens), such that the sum of probabilities across the 30 generated answers equals 1?
- What is the information included in the blocks that are retrieved from Google as context for the RAG setups?
- In Figure 1, why is the performance of MASK RAG for SQuAD-zh better than the one of the No RAG setup? This also contradicts the finding mentioned in lines 189-191 of the manuscript. I would expect that masking the gold answer with `MASK` placeholders would consistently result in performance degradation, especially given the fact that the addition of these masked tokens in this context is not explicitly learnt by the model.
- Typo: Please correct the name of the dataset in line 323: TrivialQA → TriviaQA.

Akari Asai, Zeqiu Wu, Yizhong Wang, Avirup Sil and Hannaneh Hajishirzi. 2024. Self-RAG: Learning to Retrieve, Generate, and Critique through Self-Reflection. In The Twelfth International Conference on Learning Representations, 2024.

---

### Note · Authors · 2024-11-25

I have read and agree with the venue's withdrawal policy on behalf of myself and my co-authors.